# Infectious Bovine Rhinotracheitis Post-Eradication Program in the Autonomous Province of Bolzano, Italy: A Retrospective Study on Potential Bovine Herpesvirus Type 2 Cross-Reactivity

**DOI:** 10.3390/ani13223502

**Published:** 2023-11-13

**Authors:** Astrid Bettini, Martina Stella, Francesca Precazzini, Marta Degasperi, Stefano Colorio, Alexander Tavella

**Affiliations:** Experimental Zooprophylactic Institute of the Venezie, 39100 Bolzano, Italy; abettini@izsvenezie.it (A.B.); mstella@izsvenezie.it (M.S.); fprecazzini@izsvenezie.it (F.P.); martadegasperi97@gmail.com (M.D.); scolorio@izsvenezie.it (S.C.)

**Keywords:** BoAHV1, BoAHV2, bulk milk serology, IBR

## Abstract

**Simple Summary:**

Infectious bovine rhinotracheitis is one of the diseases caused by bovine herpesvirus type 1; it affects cattle and can heavily damage the livestock industry. Several countries have adopted different prevention measures, and some of them have been declared free from infectious bovine rhinotracheitis. Our work is focused on data collected in the autonomous province of Bolzano (Italy). This province underwent an eradication program between 1991 and 2000. In 2000, the territory was declared IBR-free. Since 2001, a post-eradication program has been conducted, which includes annual milk testing. If a positive result is found, additional blood tests are performed. During the several years of the post-eradication program, positive results were detected in bulk milk and serum samples; our hypothesis is that these positive results are potentially caused by bovine herpesvirus type 2 cross-reactivity.

**Abstract:**

Bovine alphaherpesviruses, BoAHV, can cause respiratory, genital and neurological disorders. In particular, bovine alphaherpesvirus type 1 (BoAHV1) is one of the most significant ruminant pathogens worldwide and it can heavily damage the livestock industry. BoAHV1 can cause infectious bovine rhinotracheitis (IBR) along with fertility disorders. Bovine alphaherpesvirus type 2 (BoAHV2) can cause two different conditions as well: pseudo-lumpy skin disease (PSLD) and bovine herpetic mammillitis (BHM). The autonomous province of Bolzano (Italy) has adopted several strategies to control and eradicate IBR, and it was declared in 2000 to be IBR-free by the European Commission. Since 2001, a post-eradication monitoring program has overseen the serological analysis of bulk milk and, in the presence of a positive result, a follow-up examination is performed on the individual blood serum of all bovines older than 24 months that belong to bulk milk-positive herds. Despite the detection of positives in both bulk milk and serum samples, South Tyrol has been declared IBR-free, as these positives have never been confirmed through seroneutralization. Between 2014 and 2022, approximately 41,000 bulk milk (averaging 4300 samples/year) and 3229 serum samples were tested for BoAHV1. The aim of this study was to evaluate the post-eradication program for IBR with a particular focus on the potential cross-reactivity with BoAHV2; for this reason, serum samples were also tested for BoAHV2 antibodies. This study could be of great importance for those countries that submit herds to an IBR monitoring and eradication program; performing further analyses to confirm and explain false positive outcomes would increase the reliability of the obtained results.

## 1. Introduction

Bovine Alphaherpesvirus type 1 (BoAHV1) is a virus of the *Herpesviridae* family, subfamily *Alphaherpesvirinae*, genus *Varicellovirus*, known as the causative agent of infectious bovine rhinotracheitis (IBR). BoAHV1’s viral genome consists of double-stranded DNA that encodes approximately 70 proteins. Among the structural envelope proteins, two glycoproteins, gB and gE, play an essential role in immunity and pathogenesis. Furthermore, they both play major roles in virulence determination, being mainly involved in cell-to-cell transmission, virus attachment, entry and fusion [1]. BoAHV1 causes a highly contagious infectious disease in domestic and wild bovines, distributed worldwide, that presents significant differences in regional incidence and prevalence, mainly due to the geographic positioning and breeding management of the herds [2]. BoAHV1 infections lead to great economic losses, as they affect both animal health and productivity, leading to a decrease in milk production and an increase in abortion, premature culling and natural death. Furthermore, BoAHV1 infections may be financially wasteful because of strict trading restrictions in the cattle industry [3]. BoAHV1 can cause two different clinical forms: it can affect the upper respiratory tract, causing IBR, or the genital tract, leading to infectious pustular vulvovaginitis (IPV) in females and infectious balanoposthitis (IBP) in bulls, along with other fertility disorders [4]. The risk of mortality is fairly low and the disease runs as a subclinical course in most cases, facilitating secondary bacterial infections and leading to more severe respiratory diseases [5,6]. After primary infection with BoAHV1, cattle become latent carriers, with the establishment of a lifelong latency in the sensory neurons of the peripheral nervous system, which can be reactivated via several stress factors [3,7]. Viral shedding occurs mainly through respiratory, ocular and genital secretions; the infection is therefore transmitted by direct contact between animals and may occur via fresh or frozen semen from infected bulls during artificial insemination [7].

Bovine alphaherpesvirus type 2 (BoAHV2) is another virus belonging to the *Herpesviridae* family, subfamily *Alphaherpesvirinae*, genus *Simplexvirus*. Its genome is typical of most simplexviruses, and it has been shown to have several genetic similarities to that of human alphaherpesvirus 1 (HHV1). In particular, the comparison of glycoprotein B (gB) sequences of BoAHV2 and HHV1 was used as a basis for the classification of their taxonomic group [8]. BoAHV2 is associated with two main clinical forms in cattle: bovine herpetic mammillitis (BHM) characterized by vesicular, ulcerated or necrotic lesions of the teat and udder skin, and pseudo-lumpy skin disease (PSLD) characterized by generalized superficial skin lesions [9]. BoAHV2 infections have been identified in cattle worldwide. Transmission occurs, in nature, mainly between heifers and calves at first lactation, and the infection may undergo a subclinical course or be accompanied by relatively mild disease [9]. Even though BoAHV2 infections are frequently benign and lead to a self-limiting disease, they may result in a significant decrease in milk production and an increase in the susceptibility of the mammary gland to bacterial or chronic mastitis, which leads to the culling of the infected animal [10,11].

The autonomous province of Bolzano (South Tyrol, Italy) started an IBR eradication program in 1991. Vaccination with differentiating infected from vaccinated animals (DIVA) against BoAHV1 was allowed during the eradication program until December 1995 in order to decrease viral shedding and prevent clinical signs; however, it was later prohibited by provincial law in 1996, with an initial derogation for herds with extremely high prevalence and financial difficulties, and finally in 1999 for all herds in South Tyrol. Initially, vaccination was allowed to decrease shedding and spreading of the virus, but it was later forbidden to avoid false positive diagnostic results due to vaccination antibodies. Because of the average cattle lifespan of 4 years and 6 months (data from the South Tyrolean Breeding Association, 2022), no vaccinated animal or vaccination antibody titers should be detected anymore.

Control and prevention of BoAHV1 infections mainly rely on good farm management, including biosecurity measures, as well as control programs for the eradication of the virus [12]. Several countries, such as Finland, Sweden, Norway, Denmark, Austria and Switzerland, along with geographic regions, such as Bavaria (Germany), Jersey (UK), Valle d’Aosta and the autonomous province of Bolzano (Italy), have adopted such strategies and are now IBR-free. South Tyrol was declared BoAHV1-free in 2000 [13]. The South Tyrolean cattle population accounts for approximately 4200 dairy farms (Provincial Databank, 2022).

Since 2001, the BoAHV1 post-eradication monitoring program has overseen serological analysis of bulk milk using a whole-virus indirect enzyme linked immunosorbent assay (ELISA) and, in the case of a positive result, a follow-up examination via gB and gE blocking ELISAs, as well as a serum neutralization assay (SN) of the individual blood serum of all animals older than 24 months, and thus potentially lactating and belonging to a bulk-milk positive herd. The aim of this study was to evaluate the BoAHV1 post-eradication program, with particular focus on the possibility of BoAHV2 cross-reactions in bulk milk serology, as serological cross-reactivities between BoAHV1 and BoAHV2 have been previously reported [14,15,16,17]. It is noteworthy to mention that the structural homology between simplexviruses and varicelloviruses [18], which could lead to a nonspecific immune response, potentially detectable in both BoAHV1 and BoAHV2 serologic analyses.

## 2. Materials and Methods

### Bulk Milk and Serum Serological Analyses

Since 2001, all dairy farms have been subjected to yearly bulk milk serologic analyses. The samples have been collected in September of each year and prepared with a preservative agent (Azidiol, ANA.LI.TIK. e.U. Austria, Vienna, A) to ensure milk stability. The samples have then been conferred to the Istituto Zooprofilattico Sperimentale delle Venezie (IZSVe) in Bolzano for milk serology. Samples were stored under frozen conditions (−20 ± 2 °C) until analyses were performed with a commercially available ELISA for the detection of anti-BoAHV1 antibodies. Since IZSVe is a national institution and is subjected to tendering procedures every three years for all commercially available kits, during both the eradication and the post-eradication programs, different ELISA kits could have been performed for similar analyses, based on the most recent tendering-winning company. All information regarding the commercial ELISAs used is reported in Table 1. In the presence of a bulk milk-positive result, a follow-up examination was required of the individual blood serum samples of all bovines older than 24 months belonging to that herd. Blood samples were obtained from the caudal or jugular vein using vacuum blood collection tubes with a clotting activator manufactured by Vacutest Kima (Azergrande, Italy). The samples were centrifuged for 3 min at 1646× *g* to obtain the serum and then analyzed. Samples that showed a positive reaction were further tested for the detection of gB and gE antibodies with other commercial ELISAs (Table 1). All analyses were performed following manufacturers’ instructions. Furthermore, in the presence of gB-positive blood serum samples, an SN assay was used according to WOAH [3], using live BoAHV-1 strain and Madin–Darby Bovine Kidney (MDBK) cell cultures. This in-house semi-quantitative procedure oversees the detection of a cytopathogenic effect at different serum dilutions, through the use of an inverted optical microscope.

Once anti-BoAHV1 analyses were performed, the samples were tested for the detection of anti-BoAHV2 antibodies in order to rule out potential cross-reactions between the two viral strains in the serologic analyses. This procedure was only conducted during the period 2014–2017 and in 2022 due to organizational reasons.

## 3. Results

### 3.1. Bulk Milk Serology

Data collected in the nine years of the follow-up campaign are summarized in Table 2.

### 3.2. Subsequent Analyses of Serum Samples

Data from the serum serology are presented in Table 3. The discordance of results between IBR anti-BoAHV1 and gB in 2017 and 2022 was attributed to the change in the diagnostic kit used for the gB ELISA.

### 3.3. Anti-BoAHV2 Antibody Screening

During the surveillance period, blood serum samples subjected to the follow-up examination were also analyzed for the detection of anti-BoAHV2 antibodies in order to rule out an infection from this viral strain. Data are reported in Table 4.

Data on gB and anti-BoAHV2 antibodies were compared in order to evaluate how many gB-positive samples showed a positive reaction to BoAHV2 as well (Table 5). All the evaluated data were negative for an SN assay, indicating a false positive BoAHV1 and gB reaction.

### 3.4. Negative Control Group

The present study reports the data of the bulk milk serology campaigns during the period 2014–2022. Three negative control (NC) groups were established in 2014, 2017 and 2022 in order to evaluate whether bulk milk negative herds would show positive reactions for anti-BoAHV2 antibodies in blood serum samples. These herds had previously reacted negatively for anti-BoAHV1 total antibodies in bulk milk serology and the herd had a history of negative results in the past prevention campaigns. Results are reported in Table 6.

## 4. Discussion

The autonomous province of Bolzano is one of the two Italian territories, along with Region Valle d’Aosta, that has been declared BoAHV1-free [11]. The closeness and trading habits with countries that had already achieved this health status, primarily Austria and Switzerland, had initially encouraged the provincial government to implement a compulsory eradication program against BoAHV1. The program was intended to avoid trade restrictions and improve animal welfare.

The program led to the complete elimination of virus circulation in our territory, and, in the early 2000s, South Tyrol moved on to a post-eradication program that oversees the yearly bulk milk serologic analyses of all dairy herds. Furthermore, each year, blood serum samples belonging to 25% of the herds that do not confer milk undergoes serologic analyses in order to cover, over a 4-year time span, 100% of the total number of herds. During this post-eradication program, the expected outcome was the identification of all sampled cattle herds as negative to anti-BoAHV1 antibodies. However, during the time period reported in this study, unexpected bulk milk serology positive reactions have been observed but these were not confirmed by subsequent serological tests performed on individual blood samples.

Bulk milk-positive herds are subjected to a follow-up blood serum investigation on all individuals of at least 24 months of age in order to exclude a true BoAHV1 infection. Understanding the dynamics behind the unexpected bulk milk serology positive results, along with the maintenance of IBR-free health status, remain of great importance for the management and success of the post-eradication BoAHV1 program in South Tyrol.

After seeing that the initial positive bulk milk results were not confirmed by the individual blood serum serology, the authors started hypothesizing that other causes might be interfering with the diagnostic tools in use. The most plausible factor identified as an interfering agent is the infection with other herpesviruses, most likely BoAHV2 or CpHV1 [15]. Due to a lack of funds and management resources, it was not possible to focus this study on CpHV1 as well as BoAHV2; the authors concentrated their interest primarily on the potential cross-reactivity with BoAHV2, since 80% of the bovine herds delivering milk are not multispecies farms (data from the provincial database).

The positivity for anti-BoAHV1 total antibodies both in the bulk milk and blood serum serologies with positive gB was never confirmed in an SN analysis, as the cytopathogenic effect detected through it has always been determined as 100%. Furthermore, almost all blood serum BoAHV1-positive samples confirmed in a gB ELISA resulted in a BoAHV2-positive reaction as well (98.6%). This could support the hypothesis of cross-reactivity, even though a confirmatory analysis (i.e., SN) for BoAHV2 was not available for us to perform. Clinical PLSD was diagnosed in only one herd over the present study’s time period, but we do not exclude the possibility of undiagnosed subclinical cases. The authors did not focus on BHM and/or PLSD clinical cases, since the aim of this study was to highlight the potential cross-reactivity between BoAHV1 and BoAHV2 serologic analyses. Another confirming factor of a BoAHV2 infection hypothesis is the fact that all bulk milk-positive herds presented at least one BoAHV2 positive serum sample, indicating that this virus might interfere with bulk milk serology as well as individual blood serum serology. A further factor supporting this hypothesis is the phylogenetic relationship between BoAHV2 (simplexvirus) and BoAHV1 (varicellovirus). A study in human medicine [18] has highlighted cross-reactivity between HHV1 (simplexvirus) and varicella zoster virus (varicellovirus); as BoAHV1 is a varicellovirus, the authors do not exclude a similar phenomenon between the two animal viruses as well.

The detection of positive blood serum BoAHV2 samples belonging to the NC groups leads us to the hypothesis that BoAHV2 infections, depending on the stage of the infection, might interfere with BoAHV1 serologic assays. Furthermore, it is possible that dairy cows that were sampled at the follow-up examination might not have been in lactation at the time of the bulk milk sampling, and therefore might not have interfered with the bulk milk serology results but might have shown their reactivity only in individual blood serum samples. Unfortunately, it is not possible to determine whether or when this could have happened, as no data on the animals sampled for bulk milk are recorded in the provincial databank. Further analyses on anti-BoAHV2 antibodies-positive animals in the NC group, along with the culling of these animals, is not overseen by the mandatory BoAHV1 post-eradication program in South Tyrol.

The present study is based on a large set of serological data collected over a long period of time and on a large number of sampled animals. These data have highlighted how, at a first serological screening, false positives might be detected; further analyzing these samples has always led to negative results. A limitation to the present study was not being able to confirm BoAHV2 positives, since a confirmation analysis was not performed. A suggestion for further studies is to confirm all BoAHV2 positives and investigate other potential cross-reacting herpesviruses, such as CpHV1, BoAHV5 and BuAHV1.

## 5. Conclusions

Based on the evidence presented in this study and in the pre-existing literature, the authors encourage all competent authorities, especially those of territories that include grazing areas and pasture practices, to initiate an eradication and monitoring program against IBR using diagnostic tools able to differentiate between BoAHV1, gB, gE and BoAHV2 antibodies. With the knowledge of false positive reactions in BoAHV1 bulk milk serology, it is important to take into consideration potential BoAHV2 cross-reactions. In conclusion, the authors encourage performing BoAHV2 serology on individual serum samples belonging to bulk milk-positive herds.

## Figures and Tables

**Table 1 animals-13-03502-t001:** Information on commercial ELISAs used during the anti-BoAHV1 post-eradication program (2014–2022).

Analysis	Year	ELISA Kit Used	Sp/Se
Bulk Milk Serology	2014–Present	Chekit BHV-1 Tank Milk, IDEXX Laboratories, Liebefeld-Bern, Switzerland	Sp ^1^ = 98% and Se ^1^ = 90%
Serum IBR anti-BoAHV1	2014–2019	Svanovir^®^ IBR-Ab, Boehringer Ingelheim Svanova, Uppsala, Sweden	Sp ^1^ = 99.7% and Se ^1^ = 100%
	2019–Present	ID Screen^®^ IBR Indirect, ID.vet Innovative Diagnostics, Montpellier, France	Sp ^1^ = 100, Se ^1^ = 100%
Serum IBR anti-gB	2015–2017	ID Screen^®^ IBR gB Competition, ID.vet Innovative Diagnostics, Montpellier, France	Sp ^1^ = 100% and Se ^1^ = 100%
	2017–Present	Cattletype^®^ BHV 1 gB Ab, Indical Bioscience GmbH, Leipzig, Germany	Sp ^1^ = 100, Se ^1^ = 100%
Serum IBR anti-gE	2014–Present	Bovine Rhinotracheitis Virus (BHV-1) gE Antibody Test Kit, IDEXX Laboratories, Liebefeld-Bern, Switzerland	Sp ^1^ = 99.7% and Se ^1^ = 97%
Serum IBR anti-BoAHV2	2014–Present	ID Screen^®^ BHV-2 Indirect, ID.vet Innovative Diagnostics, Montpellier, France	Sp ^1^ = 100% and Se ^1^ = 100%

^1^ Sp (specificity) and Se (sensitivity), as indicated by the manufacturers.

**Table 2 animals-13-03502-t002:** Bulk milk serology results during the nine-year monitoring program.

Year	Tested Herds–Bulk Milk Samples	Positive Herds (Nb—%)	Negative Herds
2014	4900	43–0.88%	4857
2015	4712	47–1.00%	4665
2016	4650	135–2.90%	4515
2017	4760	97–2.04%	4663
2018	4484	40–0.89%	4444
2019	4686	52–1.11%	4634
2020	4384	43–0.98%	4341
2021	4270	21–0.49%	4249
2022	4224	11–0.26%	4213

**Table 3 animals-13-03502-t003:** Subsequent analyses of blood serum samples belonging to bulk milk-positive herds.

Year	Total Serum Samples	IBR Anti-BoAHV1-Positive	gB-Positive	gE-Positive	SN-Positive
2014	383 (43 herds)	6	5	0	0
2015	636 (47 herds)	2	2	0	0
2016	1216 (135 herds)	14	14	0	0
2017	903 (97 herds)	65	13	0	0
2022	91 (11 herds)	9	5	0	0

**Table 4 animals-13-03502-t004:** Frequencies and percentages of the anti-BoAHV2 antibody results of the serum samples of individual animals that were positive in the bulk milk serology.

Year	Blood Serum Samples	Anti-BoAHV1 Positive (%)	Anti-BoAHV2 Positive (%)
2014	383	6 (1.57)	144 (37.6)
2015	636	2 (0.31)	271 (42.6)
2016	1216	14 (1.51)	599 (49.25)
2017	903	65 (7.2)	449 (49.7)
2022	91	9 (9.98)	43 (47.2)

**Table 5 animals-13-03502-t005:** Frequency of the positive results in IBR anti-BoAHV1, gB and anti-BoAHV2 analyses. The last column represents percentages of samples positive to both gB and anti-BoAHV2 antibodies.

Year	Anti-BoAHV1 Positive	gB	Anti-BoAHV2	gB/BoAHV2
2014	6	5	5	100%
2015	2	2	2	100%
2016	14	14	13	92.85%
2017	65	13	13	100%
2022	9	5	9	100%

**Table 6 animals-13-03502-t006:** Frequencies and percentages of anti-BoAHV1 and anti-BoAHV2 antibody results in the negative control groups.

Year	Tested Animals	Anti-BoAHV1 Positive (%)	Anti-BoAHV2 Positive (%)
2014	211	0 (0)	34 (16.11)
2017	368	0 (0)	99 (26.9)
2022	91	0 (0)	4 (4.4)

## Data Availability

All data supporting the present study is reported in this manuscript.

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
