# Peer review of "Infectious Bovine Rhinotracheitis Post-Eradication Program in the Autonomous Province of Bolzano, Italy: A Retrospective Study on Potential Bovine Herpesvirus Type 2 Cross-Reactivity"

_animals, 2023, doi:10.3390/ani13223502_

Round 1

Reviewer 1 Report

Comments and Suggestions for Authors

Dear Authors,

The proposal of manuscript is very good, the importance to verify serological cross reactions between Bovine Herpesvirus species to avoid wrong interpretations is necessary on a National Eradication Program. However some topics must be clarify:

Material and Methods:

Line 128: the BoHV-1 strain used in the SN assay was inactivated? Why have you used two cell lines for BoHV-1? Have you found differences of BoHV-1 titers?

Results:

Table 2 (Line 140): suggestion to change non-negative for positive in order to avoid confusion when read.

Table 3 (Line 143):

- same suggestion on Table 2.

- Question: why SN assay against BoHV-1 was seronegative if have you found Antibodies anti-BoHV-1 in ELISA test? Which BoHV-1 titer was determined to SN assay?

Table 4 (Line 149): same Table 2. 

Table 5 (Line 155-156): is there some possibility of cross reaction between gB and BoHV-2 considering that 100% of agreement of both ELISAs?

Question about BoHV ELISA tests: on the validation test kit for customer it was verified about cross reaction with BoHV-5? BoHV-1 and BoHV-5 there are 95% of homology, so to built a recombinant protein is very difficult for ELISA kit. However between BoHV-1 and BoHV-2 is little more difficult due to difference of genus.

Suggestion to include more references about cross reativity between bovine herpesvirus, including BoHV-5 and BoHV-1.

Comments on the Quality of English Language

Dear Authors,

We would like only to review the Verb tense. Some paragraphs the verb tense is written in Present or Past tense, mainly Material and Methods.

Examples: Line 108, 130.

Reviewer 2 Report

Comments and Suggestions for Authors

Dear Editor,

  The manuscript entitled “Infectious Bovine Rhinotracheitis post eradication program in the Autonomous Province of Bolzano –Italy: A retrospective study on Bovine Herpes Virus Type-2 cross-reactivity” Bettini et al. describes the post-eradication program conducted in the autonomous province of Bolzano (Italy) against Infectious Bovine Rhinotracheitis (IBR), between 2014 and 2022. In particular, 41000 bulk milk and 3229 serum samples were tested for BoAHV1, with a particular focus on a potential cross-reactivity with BoAHV2. For this reason, serum samples for BoHV-2 antibodies were also tested. The results evidenced: (i) a positive of gB/BoAHV2 from 92.85% (2016) to 100% (2022); (ii) a positive to anti BoAHV-2 non negative from 4% (2022) to 26.9% (2017).       

 I think this manuscript can be worth publishing if the following points are inserted:

 General comments

  1) The acronyms of Bovine alphaherpesvirus 1 (BoAHV1) and Bovine alphaherpesvirus 2 (BoAHV2) should be used throughout the manuscript by the latest International Committee of Taxonomy of Viruses (ICTV, 2022).

2) The discussion section mentions BoAHV2 and CpHV1 as potentially cross-reactive viruses. It is suggested that the authors also discuss the results concerning CpHV1, explaining why investigations were not carried out about this virus being an agent responsible for serological cross-reactivity. In addition, with regard to serological cross-reactivities with BoAHV1, it should be explained whether buffaloes in the province of Bolzano are receptive to bubaline alphaherpesvirus 1 (BuAHV1), which is also responsible for serological cross-reactivity with BoAHV1.

3) In the Discussion Section it should be explained whether positive serological reactions to BoAHV2 were associated with clinical Pseudo-Lumpy skin disease (PLSD). It would also be necessary to know whether the positive animals in the negative groups were culled and whether investigations for the virus were carried out in the target organs.

4) In addition, clarification is requested as to whether serological positivities obtained for BoAHV2 in ELISA were confirmed by viral serum neutralization tests or by plaque reduction.

5) In the Discussion Section, include the study’s limitations and whether future studies would be desirable.

 Specific Comments

 1) Page 2, Lines 60-63: citation no. 7 is inappropriate. Please change it to a more suitable one.

2) Page 3, line 111: Please, clarify what is meant by "bulk milk non-negative result". Why is not it defined as positive? And what is the difference between a non-negative and a positive result? Clarify this fact.

3) Page 3. Line 128: Please, change "Madin-Darby Bovine Kidney (MDCK)" into "Madin-Darby Bovine Kidney (MDBK)".

Comments on the Quality of English Language

Reviewer 3 Report

Comments and Suggestions for Authors

General comments

The term “non-negative results” instead of “positive results” is used throughout the manuscript. Even though I get an idea of this choice, I think it is sometimes inappropriate and confusing for readers interested in diagnostic test but not familiar with BoHV1. In the same way, the term “false non-negative results” becomes unclear. I strongly recommend to use as much as possible “positive results” and “false positive results”, and the use of “non-negative results” for results obtained by screening tests performed on bulk milk samples.

Simple summary

Line 8: “Infectious Bovine Rhinotracheitis is one of the diseases caused by bovine herpesvirus type 1;”

Line 9: “countries”, without a capital letter

Line 10: “are declared free from Infectious Bovine Rhinotracheitis”.

Line 13: “…has been conducted…”

Lines 16-17: What do you mean? that the non-negative results could be potentially related to bovine herpesvirus type 2 cross-reactivity? Please rephrase the sentence.

Abstract

The abstract needs to be revisited.

Lines 18-24: the pathologies induced by the two viruses (BoHV1 and 2) should go to the Introduction section.

Lines 29-30: the sentence is confusing: both milk and serum samples are positive, although the status is determined as IBR free…May be the analytical procedure needs to be explained briefly for a clear understanding (positive results with screening and gB tests and negative results by confirmatory tests - gE and SN).

Lines 33-36: the authors make a recommendation for countries with ongoing control program for IBR, but without any information on the results obtained in this study.

Introduction

Line 40: change “Herpes Virus” by “herpesvirus” in all sections of the manuscript, including in the abstract.

Line 42: double-stranded

Line 59: “…which can be reactivated…”

Line 72: start the sentence by “Transmission occurs…”

Line 80: Differentiating Infected from Vaccinated Animals (DIVA)

Line 81: “…in order to decrease virus shedding and prevent clinical signs.”

Line 87: “…mainly rely on…”

Line 89: “South Tyrol has been declared BoHV-1-free in 2000 [13]”. Move this sentence at the end of the next one.

Line 89: “Several countries…”. Moreover, write “countries” without a capital letter. Check this point throughout the manuscript.

Line 90: “…along with geographic regions…”

Lines 95-96: “…foresees serological analysis on bulk milk using whole-virus indirect ELISA and, in case of positive result, a follow-up examination by gB and gE blocking ELISAs as well as serum neutralization assay - SN) on…”

Line 102: “It is noteworthy to mention…”

Materials and Methods

Line 108: “…have been subjected to…

Line 110: may be “stability” rather than “durability”

Lines 108-133: use past simple rather than present.

Line 114: anti-BoHV1 antibodies

Line 116: change “because of this” by “Consequently”

Lines 116-117: “…different ELISA kits could have been performed for similar analyses,...”

Line 128: BoHV1-inactivated? Please clarify.

Lines 132-133: This procedure was only conducted during the period 2014-2017 and in 2022.

Results

General comment: the data depicted in the different tables are poorly described in the text. Please make sentences to present the main results in more details.

Line 141: title of paragraph 3.2 “Follow up examination” is imprecise. It could be changed by “Serum serology” or “Subsequent analyses of serum samples”, in accordance with title of the first paragraph (Bulk milk serology)

Line 142: “Data” is a plural word and should be conjugated accordingly.

Paragraph 3.2, table 3: the discordance of results between indirect and gB ELISA tests appears to be higher in 2017 and 2022, compared with previous years, which coincides with the use of different gB kits showing similar performance (sensitivity and specificity). Please could you comment this point.

Lines 157-158: “…campaigns during the period 2014-2022.”

Discussion

Lines 168-175: please move this paragraph in the introduction.

Lines 173-174: “…to decrease shedding and spread of the virus,…”

Lines 174-175: circulating vaccination antibodies? Please rewrite.

Lines 176-184: this second paragraph should also be moved in the introduction section because it briefly describes the workflow of the analytical procedure used in the post-eradication program and suggests that non-negative results in bulk milk analyses may correspond to false-positive reactions.

Line 184: “…and not confirmed by subsequent serological tests performed on individual blood samples.”

Lines 190-196: the authors hypothesized that the circulation of alphaherpesviruses antigenically related to BoHV1 could be responsible for false-positive reactions in bulk milk testing. Among these BoHV1-related viruses, two viruses are mentioned, including BoHV2 and CpHV1. However, bubaline alphaherpesvirus 1 (BuHV1) is another BoHV1-related virus. It has been detected in water buffaloes in Italy, and can infect cattle. Like BoHV2, cross reaction between BoHV1 and BuHV1 on serological tests can occur. Particularly, blood samples from BuHV1-infected animals can yield conflicting results in BoHV1 ELISA tests (positive and negative in gB and gE tests, respectively). Since buffalo and cattle are often raised together in mixed farms in Italy, which creates favorable conditions for cross-infections, it cannot be excluded that few cases of non-negative results in bulk milk testing of dairy herds occur after trade exchange of cattle infected with BuHV1. This question should be addressed in the section discussion.

Lines 197-204: the results of this study corroborate those of previous studies showing that BoHV2 may interfere with BoHV1 diagnosis. However, previous relevant works and their comparison with results of this study are not discussed. A paragraph dedicated to this aspect would benefit the readers.

Line 211: change “…might and might not interfere with BoHV-1 serologic assays.” by “…might interfere with BoHV-1 serologic assays.”.

Table 1

Please indicate the references to whom the performances of the kits refer.

Table 2

Change Nr by Nb in the second column.

Table 4

Combine tables 3 and 4 to a single table. Please add a hyphen like this “anti-BoHV1” or “anti-BoHV2”. Check throughout the text. The title of table 4 should be rewritten.

Table 5

Line 155: “IBR anti-BoHV2 analyses” is incorrect. Change to “anti-BoHV2 analyses”.

Comments on the Quality of English Language

The written English could be improved from a good rereading and rephrasing

Round 2

Reviewer 1 Report

Comments and Suggestions for Authors

Dear Authors,

Thank you very much to attendant my suggestions. However about SN assay (Lines 135 to 138) is still have some mistake of interpretation about the technique. For SN assay is necessary a LIVE BoAHV-1 in order to observe presence or absence of cytopathic effec on MDBK cells. If animal is free for BoAHV-1 antibodies, the LiveBoAHV-1 linked with MDBK receptor cells and promote cytopathic effect. In contrast, if animal has antibodies, it could neutralize of Live BoAHV-1 and MDBK cells are fine. Please clarify.

Concerning about BoAHV-5, the authors understood my question due to response in the end of Discussion (Lines 246 to 252).

Author Response

Thank you for your correction. We modified the manuscript accordingly. We apologize for the mistake.

Kind regards.